**Data availability statement:** All relevant data for this study are publicly available from the

# Assessing the generalization capabilities of TCR binding predictors via peptide distance analysis

**Leonardo V. Castorina**[1,2]*, **Filippo Grazioli**[2], **Pierre Machart**[2], **Anja Mösch**[2], **Federico Errica**[2]

**1** School of Informatics, University of Edinburgh, Edinburgh, United Kingdom, **2** NEC Laboratories Europe, Heidelberg, Germany

* leonardo.castorina@ed.ac.uk

## Abstract

Understanding the interaction between T Cell Receptors (TCRs) and peptide-bound Major Histocompatibility Complexes (pMHCs) is crucial for comprehending immune responses and developing targeted immunotherapies. While recent machine learning (ML) models show remarkable success in predicting TCR-pMHC binding within training data, these models often fail to generalize to peptides outside their training distributions, raising concerns about their applicability in therapeutic settings. Understanding and improving the generalization of these models is therefore critical to ensure real-world applications. To address this issue, we evaluate the effect of the distance between training and testing peptide distributions on ML model empirical risk assessments, using sequence-based and 3D structure-based distance metrics. In our analysis we use several state-of-the-art models for TCR-peptide binding prediction: Attentive Variational Information Bottleneck (AVIB), NetTCR-2.0 and -2.2, and ERGO II (pre-trained autoencoder) and ERGO II (LSTM). In this work, we introduce a novel approach for assessing the generalization capabilities of TCR binding predictors: the Distance Split (DS) algorithm. The DS algorithm controls the distance between training and testing peptides based on both sequence and structure, allowing for a more nuanced evaluation of model performance. We show that lower 3D shape similarity between training and test peptides is associated with a harder out-of-distribution task definition, which is more interesting when measuring the ability to generalize to unseen peptides. However, we observe the opposite effect when splitting using sequence-based similarity. These findings highlight the importance of using a distance-based splitting approach to benchmark models. This could then be used to estimate a confidence score on predictions on novel and unseen peptides, based on how different they are from the training ones. Additionally, our results may hint that employing 3D shape to complement sequence information could improve the accuracy of TCR-pMHC binding predictors.

GitHub repository (https://github.com/nec-research/peptide-distance-analysis).

**Funding:** Funding for this project was provided by NEC Labs Europe GmbH.

**Competing interests:** Funding for this project was provided by NEC Labs Europe GmbH.

## Introduction

The immune system has evolved to recognize different pathogens, such as viruses, which enter cells and exploit their resources for replication. The immune response depends on how well the system can distinguish between healthy and infected/aberrant cells. It does so by leveraging specific molecules on the cell surface called Major Histocompatibility Complexes (MHC). Class I or class II MHCs are presented depending on the cell type [1–3]. An antigenic peptide is displayed by the MHC forming a peptide-bound MHC (pMHC). These peptides are derived from the proteasome, a complex of protease enzymes in cells which break down proteins into short peptides [4].

In infected cells, the presented peptides may be derived from viral proteins, whereas in healthy cells, peptides are derived from housekeeping proteins [5]. T cells can recognize viral peptides in the pMHC by binding it with the T Cell Receptors (TCRs), leading to an immune response [4]. TCRs are able to recognize many different peptides via diverse sequences at the variable regions of their $\alpha$ and $\beta$ chains [6]. TCR binding to the pMHC primarily occurs at the Complementarity-Determining Regions (CDRs). The peptide recognition is primarily mediated by the CDR3. The CDR3-$\alpha$ is derived from the alleles of the V and J genes; the D gene, in addition to the V and J, is involved in shaping the sequence and structure of the CDR3-$\beta$, [7,8]. These alleles can be recombined extensively, ensuring a high TCR repertoire diversity and allowing for a broad T cell-based immune response [9]. The exposure of a naive T cell to an antigen leads to its activation and to the development of a memory T cell population with the same TCR. This allows for a long-lasting immune response [10,11]. A visual overview of the TCR-pMHC complex is presented in the S1 File.

Immunotherapies targeting cancer or viral infections use binding specificity to activate T cells, enhancing the immune system response. Binding specificity ensures that T cell binding occurs uniquely on the target, to kill say cancerous or infected cells, and avoid healthy cells. Research in immunotherapy covers two main categories: adoptive T cell therapy and T cell inducing vaccines. In adoptive T cell therapy, cancer patients receive specific T cells targeting and destroying the tumour [12,13]. T cell inducing vaccines leverage antigenic peptides or other classes of antigens to trigger specific T cell development against pathogens, such as viruses [14,15].

Immunotherapy design requires an in-depth understanding of the biochemical interactions between TCRs and pMHCs to accurately stimulate the immune system. The development of computational models that predict whether TCRs and pMHCs interact would allow for a faster *in silico* screening of sequences and consequent design of TCR sequences that bind specific target pMHCs. Recent advancements in machine learning (ML) lead to the development of TCR binding predictors [16–22]. The data input of these models consists of tuples of short amino acid sequences such as (peptide, CDR3-$\beta$) or (peptide, CDR3-$\beta$, CDR3-$\alpha$, MHC). The task is usually formalized as a binary classification between binding or non-binding pairs.

Several computational studies on TCR binding prediction employ data from the Immune Epitope Database (IEDB) [23], VDJdb [24] and McPAS-TCR [25]. These databases mainly contain CDR3-$\beta$ sequences, but often lack information on CDR3-$\alpha$. Additionally, public TCR-pMHC interaction datasets are often limited in diversity and size. They also present sequence bias towards commonly studied viruses or binding/non-binding pairs bias from experiment setups. Furthermore, laboratory methods that validate the TCR-pMHC interactions, such as surface plasmon resonance, titration calorimetry and fluorescence anisotropy, are usually resource-intensive and time-consuming, hindering the creation of larger datasets [26].

Generally, machine learning (ML) TCR binding predictors achieve high test performance when evaluated on test sets originating from the same source as the training set. However, various studies [27–29] have shown that these methods exhibit weak cross-dataset generalization, meaning, models performance is significantly lower when training and test samples come from different distributions. Furthermore, [27] investigated the effect of different splitting techniques on the TChard dataset, which combines samples from IEDB, VDJdb and McPAS-TCR. They also investigated the effect of considering negative samples from wet lab assays and from random shuffling of the positive tuples. They observed that, when using a "vanilla" Random Split (RS) with negative samples from wet lab assays, ML-based models achieve an estimated area under the receiver operator characteristic (AUROC) larger than 95% on the test set. In this setting, the sets of CDR3 sequences in positive and negative samples are disjoint. This allows the models to memorize whether a CDR3 sequence was observed in either the negative or positive samples at training time. [27] then showed that a simple countermeasure to this problem consists in creating negative samples by randomly shuffling the positive tuples. In this setting, when the RS is employed, models achieve an AUROC score between 70% and 80%. Nevertheless, using the RS, peptides and CDR3 sequences may appear in both the training and test sets, leading to inflated estimates of the real-world model generalization capabilities.

Hence, to approximately test model performance for unseen peptides, they propose an alternative splitting method, named Hard Split (HS). The HS ensures that test peptides are never observed at training time. Therefore, the peptide sequences in the training, validation and test sets are unique to their respective sets. When using the HS and negative samples are obtained via random shuffling, the model performs barely better than random guesses, with an AUROC smaller than 55%.

To ensure real-world applicability, the HS, as shown by [27], aims at evaluating models' generalization abilities under the toughest setup possible. In fact, new peptides arising from new pathogens may differ significantly from those used at training time. ML models can only be safely employed for broad real-world applications if they show *sufficient* generalization to unseen sequences.

In practice, however, RS and HS represent two extremes of a wider spectrum of data splitting options. Assessing other splitting options can provide further insights on the robustness of ML models.

Our work is driven by the following research question: can we estimate the performance of ML models given an unseen test peptide and its distance from the training ones? We hypothesize that, as the difference in distributions between training and test sets increases, the prediction task becomes gradually harder. Knowing how well ML predictors generalize to an increasing shift in distributions between training and test sets would provide a tangible metric to measure when it is appropriate and to which extent we can employ these methods in the real world.

For this reason, in this work we introduce a new dataset splitting algorithm called *Distance Split (DS)*. Analogously to the HS, peptides placed in the training set are absent from the test set and vice versa. However, given a distance metric, we control the distances between sequences in the training and in the test sets. We increase such distance to test performance on peptides that are "further apart" from the ones seen by the model at training time, making the task increasingly harder.

Previous efforts have predominantly relied on sequence-based metrics, such as Levenshtein distance and BLOSUM substitution matrices, to measure similarity between peptides [30–32]. However, sequence metrics may not fully capture the structural aspects of TCR-peptide interactions, which are essential for accurate binding predictions. The integration of 3D structural

information, such as Root Mean Square Deviation (RMSD) between peptides, could offer a more comprehensive approach to evaluating model performance on unseen peptides. Therefore, we explore different distance metrics, i.e., sequence metrics such as Levenshtein and BLOSUM [33], as well as shape metrics such as RMSD [34], using the predicted structures of peptides. As opposed to binning the HS based on distance, the DS allows for control over the median distance of the peptides in the test and validation.

We show that the increased peptide distance between the training and test split directly correlates with the models performances. In particular, increased RMSD (shape) distance between training and test peptides leads to decreased performance. Surprisingly, for BLOSUM (sequence) distance we see the inverse relationship. In real-world scenarios, when predictions on new unseen viral epitopes are required, we believe that leveraging RMSD between the 3D structures of the training and test peptides could serve as a valuable indicator of model reliability, with higher RMSD implying increased prediction uncertainty. Conversely, the inverse relationship observed with BLOSUM distance suggests that incorporating sequence-diverse training data may actually improve model generalization to novel peptides. Thus, a combination of structural and sequence-based metrics could provide a balanced approach, enhancing both the robustness and reliability of TCR-pMHC binding predictions.

## Materials and methods

### Dataset creation

With the goal of vaccine development against viruses, we select a viral subset of the VDJdb dataset, focusing on human host and MHC class I [35]. We omit MHC class II due to the small number of available samples. The dataset includes peptides from SARS-CoV-2, Influenza A, Human Immunodeficiency Virus (HIV), Hepatitis C Virus (HCV) and Cytomegalovirus (CMV). We discard data points which do not include both the CDR3-$\beta$ and the peptide. The dataset contains 52 unique MHC A and 1 MHC B alleles, 16,504 unique CDR3-$\alpha$ sequences, 28,831 unique CDR3-$\beta$ sequences and 757 unique peptides, for a total of 34,415 binding samples. To generate non-binding (i.e., negative) samples, we randomly shuffle the available (peptide, CDR3-$\beta$) pairs. This process is commonly employed in the literature [19,36] and leverages the hypothesis that random pairs will most likely not bind. To create a balanced dataset, we randomly generate 36,641 samples of non-binding combinations of CDR3-$\beta$ and peptide sequences, to increase the total number of data points to 65,946. In this study, the data consists of pairs of (peptide, CDR3-$\beta$) and a binding/non-binding label.

**Obtaining 3D structures of peptides.** The VDJdb dataset contains information on the primary structure of proteins, i.e., the sequence of amino acids. In reality, these sequences exist as 3D shapes and as part of a complex [35]. However, public datasets like VDJDB lack the 3D shapes of peptides. We use ESMFold, a light-weight sequence-to-shape Language Model [37] and its online API[1] to generate 3D structures for the peptide sequences. For structures that gave errors, we use OmegaFold [38], a similar sequence-to-shape Language Model. We use the shapes to calculate the 3D distance between peptides, as described in the following section.

### The distance split algorithm

Given a distance metric, the goal of the Distance Split (DS) algorithm is to create data splits of the available (peptide, CDR3-$\beta$) samples, enforcing a specified median peptide distance between training and test (and validation).

---

[1] https://esmatlas.com/resources?action=fold.

As distance metrics, we use Levenshtein, BLOSUM and RMSD. The Levenshtein distance is a string edit distance measuring the minimum number of edits required to change one peptide sequence into another [39]. The BLOSUM distance is a substitution matrix-based metric that quantifies the evolutionary similarity between two peptide sequences by comparing their amino acid global alignments [40]. The RMSD distance is a 3D structural metric that measures the average deviation between the atomic positions of two superimposed peptide structures, providing an estimate of their conformational similarity. We calculated the RMSD shape distance using PyMol [34]. PyMol aligns any pair of peptide 3D structures computes the RMSD between C$\alpha$ of the two structures. The C$\alpha$ atoms are the backbone carbon atoms in each amino acid, which are commonly used in RMSD calculations to provide a simplified yet accurate representation of the overall peptide structure. We focus on C$\alpha$ atoms to avoid over-penalizing minor deviations in side chains, as protein folding models can generate physically impossible conformations that would otherwise lead to inflated RMSD values [41].

For each metric (Levenshtein, BLOSUM, RMSD), we calculate the pairwise distances between all 757 peptides, resulting in three distance matrices: $\mathbf{M}_{Levenshtein}$, $\mathbf{M}_{BLOSUM}$ $\mathbf{M}_{RMSD}$ $\in \mathbb{R}^{757 \times 757}$. For each distance matrix, we then calculate the row-wise median $\mathbf{m}_{med} \in \mathbb{R}^{757}$. Each value of $\mathbf{m}_{med}$ corresponds to the median distance between a given peptide and all other peptides. We then select three bin ranges over the cumulative distribution of the median distances of $\mathbf{m}_{med}$, defined by a lower and an upper bound $(b_l, b_u)$ distance, i.e., (0,33), (33,66) and (66,100). Given the median vector $\mathbf{m}_{med}$, for each interval, we compute the distances corresponding to the lower and upper percentiles, i.e., $d_l$ and $d_u$, for $b_l$ and $b_u$, respectively. We then filter out all the peptides whose median distance falls outside the interval of interest. The sets of peptides that fall inside this percentile interval are then sampled for test and validation. For each training-validation-test split, we use a 90-5-5 ratio. The DS algorithm iteratively samples the validation and test sets based on this ratio and the total available data. To guide this process, we calculate the expected number of data points for each split $N_{train}$, $N_{test}$, $N_{validation}$, which we refer to as the test and validation budget.

We select the test samples by iteratively sampling peptides from the specified percentile interval. In each iteration, we randomly choose a peptide from this set, and move all corresponding (peptide, CDR3-$\beta$) pairs to the test set. This process continues until the target number of test samples (test budget) is reached. Peptides that were not selected are used to form the validation set, following the same iterative procedure. Then, the remaining peptides are assigned to the training set.

Additionally, we enforce a minimum and maximum count for peptides to be included in the validation and test sets. This is meant to constrain the peptide diversity in the set. In this work, we use a minimum count of 5 maximum count of 5,000. The complete algorithm for the DS is available in 1.

Using the DS, as well as the RS and HS, we generate training, test and validation splits of the available data points for all distance metrics, $\mathbf{M}_{Levenshtein}$, $\mathbf{M}_{BLOSUM}$ and $\mathbf{M}_{RMSD}$. As shown in S4 Fig, the average number of unique peptides is roughly the same across splits.

## TCR-peptide interaction prediction models

We select 5 state-of-the-art models for TCR-peptide interaction prediction: Attentive Variational Information Bottleneck (AVIB) [27], NetTCR-2.0 [36], NetTCR-2.2 [42], ERGO II (with pre-trained TCR autoencoder), and ERGO II-LSTM [16] (see details in S1 File, Sect 2).

**Algorithm 1.    Distance Split (DS) algorithm based on peptide distances within specified percentile bounds.**

**Input:**   Dataset of peptide-TCR pairs $\mathcal{D} = \{(pep_i, TCR_i)\}_{i=1}^{N}$;
Peptide distance metric (e.g., BLOSUM or RMSD);
Training split ratio s (fraction of data for training);
Lower and upper percentile bounds $(b_l, b_u)$ over the distance distribution;
Leeway factor l: flexibility when the exact sample budget cannot be met. If the number of available samples falls below the target by up to $l \times N$, the algorithm will accept the smaller set.
**Output:** Training, validation, and test sets.
**1) Compute unique peptides and counts:**
• Extract the set of unique peptides $\mathcal{P}$ from $\mathcal{D}$.
• $\forall$ $pep \in \mathcal{P}$, count the number of (pep,TCR) pairs and store in CountMap(pep).

**2) Calculate peptide distances:**

• Compute the distance matrix $\mathbf{M}_{dist}$ between peptides using the chosen method.
• Calculate the median distance $\mathbf{m}_{med}$ for each peptide from $\mathbf{M}_{dist}$.

**3) Select peptides within percentile bounds:**

• Determine $d_l$ and $d_u$ distances corresponding to percentiles $b_l$ and $b_u$ from $\mathbf{m}_{med}$.
• Select peptides $\mathcal{P}^* = \{pep \in \mathcal{P} \mid \mathbf{m}_{med}(pep) \in [d_l, d_u]\}$.

**4) Determine split sizes:**

• Calculate total sample count: $N_{total} = \sum_{pep \in \mathcal{P}} CountMap(pep)$.
• Compute training sample count: $N_{train} = s \times N_{total}$.
• Remaining samples: $N_{remaining} = N_{total} - N_{train}$.
• Set test and validation counts: $N_{test} = N_{validation} = N_{remaining}/2$.

**5) Initialize sets:**

• $\mathcal{P}_{train} \leftarrow \mathcal{P}$, $\mathcal{P}_{test} \leftarrow \{\}$, $\mathcal{P}_{val} \leftarrow \{\}$.
• Counters: $t_{test} \leftarrow 0$, $t_{val} \leftarrow 0$.

**6) Sample peptides for test and validation sets:**
**for** *each set in {test, validation}* **do**
 **while** *counter t < N × (1–l)* **do**
 Randomly sample pep from $\mathcal{P}^*$ not already selected.
 **if** $t + CountMap(pep) \leq (1 + l) \times N$ **then**
 Add pep to the current set.
 Remove pep from $\mathcal{P}_{train}$ and $\mathcal{P}^*$.
 Update counter: $t \leftarrow t + CountMap(pep)$.
 **end**
 **end**
**end**

**7) Construct final training, validation and test datasets:**

• $\mathcal{D}_{train} = \{(pep, TCR)_i\}$, where $pep \in \mathcal{P}_{train}$; $\mathcal{D}_{val} = \{(pep, TCR)_i\}$, where $pep \in \mathcal{P}_{val}$; $\mathcal{D}_{test} = \{(pep, TCR)_i\}$, where $pep \in \mathcal{P}_{test}$.

With the exception of ERGO, all models encode the TCR$\beta$ and peptide sequences using the BLOSUM substitution matrix [40].

In AVIB, the TCR$\beta$ and Peptide encoders estimate Gaussian posteriors over a latent space. The various posteriors are then combined using an attention mechanism to estimate a single multi-sequence posterior distribution, which is then used to estimate the binding probability [27]. In NetTCR-2.0, the TCR$\beta$ and peptide encoders are encoded using several convolutional and global max pooling layers. The encoding is then concatenated and passed through two fully connected layers [36]. NetTCR-2.2 introduces a dropout layer before the fully connected layers, ReLU over Sigmoid, and 64 units rather than 32 for the fully connected layer [42]. In the ERGO II model, the TCR$\beta$ is encoded using a pre-trained autoencoder. ERGO II-LSTM encodes the peptide using an LSTM encoder. The encodings are then passed to two fully connected layers [16].

## Results

We compute the pairwise distance between all peptides using Levenshtein, BLOSUM and RMSD distance for the viral VDJdb dataset. Descriptive statistics of the metrics are summarized in Table 1 and the median visualized in Fig 1. The peptides count distribution is heavily skewed towards a count of less than 10 instances, with several outliers, as shown by the high standard deviation. The median RMSD has a bimodal distribution with peaks at $\approx$ 0.0 and $\approx$ 0.5 Å. The median BLOSUM is normally distributed with most values ranging from 50 to 100. Finally, the median Levenshtein distance is $\approx$ 8, and shows very little distribution of values.

Then, we calculate the correlation between these distances in Fig 2. Levenshtein and BLOSUM distances are very significantly correlated (Spearman $\rho$: 0.32, p-value: 0), as expected since BLOSUM distance uses a sequence alignment. However, additional physico-chemical information is implicitly accounted for in BLOSUM, which considers evolutionary substitution patterns of amino acids [40]. This inclusion captures the likelihood of biologically relevant substitutions based on factors like size, charge, and hydrophobicity, which are not reflected in simple sequence-based metrics like Levenshtein distance. We observe no correlation between RMSD and sequence-based metrics.

For our experiments, we use five state-of-the-art deep learning models for TCR-peptide interaction prediction: NetTCR-2.0 [36], NetTCR-2.2 [42], AVIB [19], ERGO II and ERGO II-LSTM [16]. We evaluate how increasing the distance between the training and test set affects their performance. We train and test all models with RS, HS as baselines and and DS. For the

**Table 1. Descriptive statistics of peptide counts, as well as Levenshtein, BLOSUM, and RMSD distances for the viral VDJdb dataset. Levenshtein and BLOSUM are sequence distance metrics. RMSD refers to the average 3D distance between peptides.**

| Metric | Average | Median | Stdev |
|---|---|---|---|
| Peptide (counts) | 45 | 2 | 576 |
| Levenshtein | 8.29 | 8.25 | 0.60 |
| BLOSUM | 85.04 | 71.10 | 28.97 |
| RMSD (Å) | 0.46 | 0.36 | 0.27 |

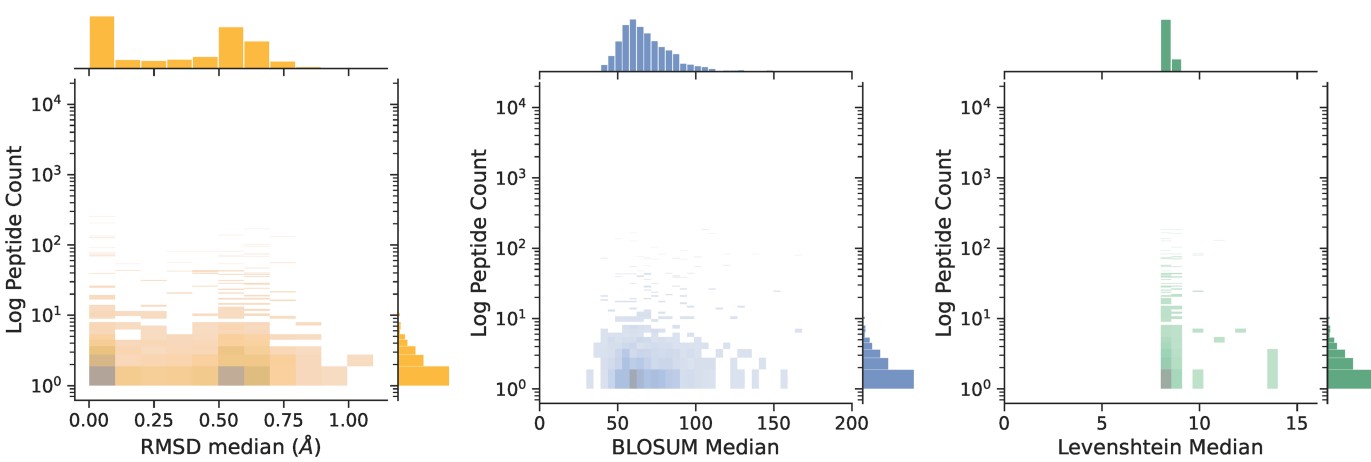

**Fig 1. Distribution of peptide count against median distance of the peptide against all other peptides.** Distances are RMSD (3D shape), BLOSUM (sequence), and Levenshtein (sequence), respectively.

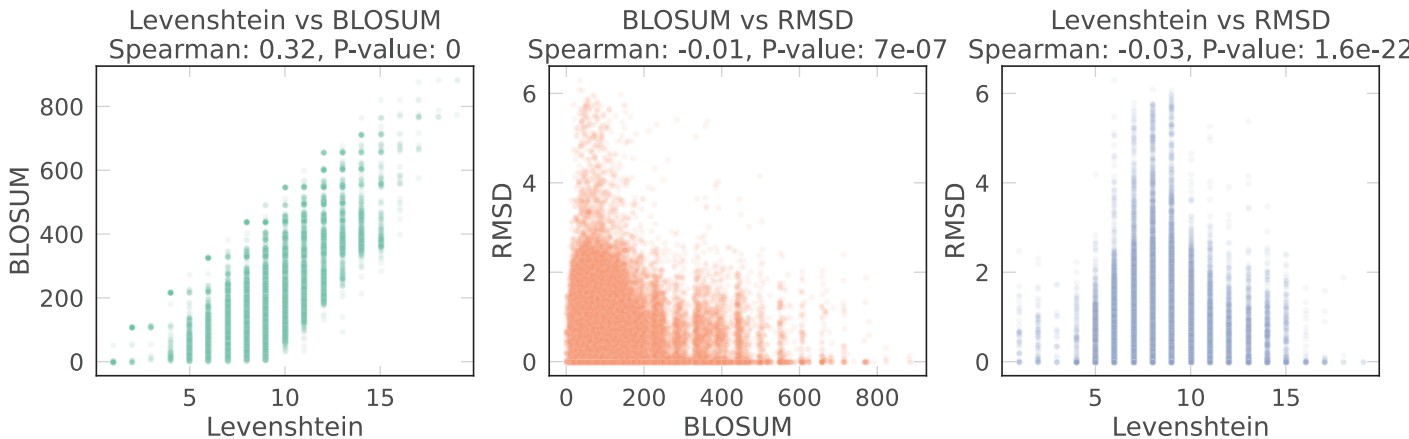

**Fig 2. Correlations between peptides distance metrics in the Viral VDJDB dataset.** Levenshtein and BLOSUM are sequence-based distances, while RMSD is a shape-based distance. There is a positive correlation between sequence distances and no correlation between shape and sequence distances.

DS, we use lower and upper bound pairs over the cumulative distribution of the median distances: (0, 33), (33, 66) and (66, 100). We repeat each split with 5 different random seeds and the average results and 95% confidence intervals are shown in Fig 3. More detailed results with Levenshtein DS, overall and per-peptide metrics are available in S1 File Sect 2.

As shown in Fig 3, AVIB and NetTCR-2.2 perform similarly in the sequence and shape DS splits, followed by NetTCR-2.0 and both ERGO II models. As expected, all models perform best in the RS, as peptides in the training set also appear in the test set. On the other hand, performance on the HS is significantly lower compared to the RS.

Regarding the DS splits, as the median distance between the training and test sets increases, model performance worsens in RMSD-based DS splits, as shown by significant Spearman correlations ($p < 0.05$) for NetTCR-2.2, NetTCR-2.0, and AVIB, all within a 95% confidence interval. In contrast, we observe the opposite trend for BLOSUM-based DS splits, where models perform better as the median BLOSUM distance between the training and test sets increases, with all p-values below 0.05. In general, the highest performance in the DS splits is slightly higher in the RMSD (0,33) bin, than in the BLOSUM (66,100) bin, except for ERGO II (LSTM). There is generally no significant effect on the performance trend when using Levenshtein-based distance split, except for ERGO II (LSTM) which shows a similar trend to BLOSUM (see S14 Fig).

## Discussion

The TCR-peptide/pMHC interaction prediction problem presents evaluation challenges [27]. When data is randomly split, models can achieve over-optimistic test performance, as test sequences can be observed at training time.

As a solution, [27] proposed the Hard Split (HS), which consists in randomly sampling test peptides and moving all instances of that peptide to the test set. The HS ensures that test peptides are never seen at training time, thus simulating what happens in the real world, for example, when new peptides arise from new pathogens.

This raises the question of whether it is possible to infer how well a model will generalize to an unseen peptide, by knowing how different the novel peptide is from those seen at training time.

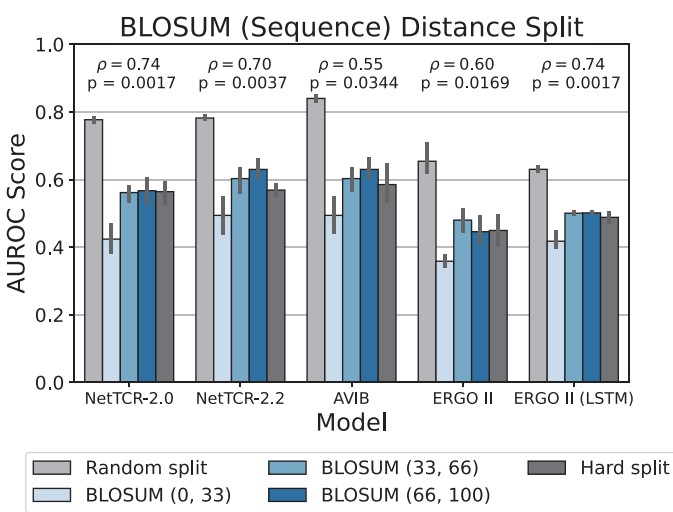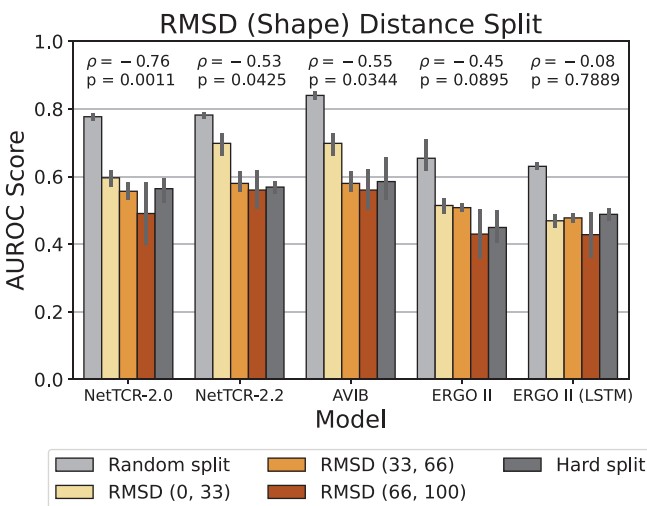

**Fig 3. AUROC Scores for TCR's CDR3β-peptide binding prediction for models trained and tested using various splitting techniques.** The commonly used *Random Split* allows test peptides to be observed also during training. In the *Hard Split*, peptides are exclusively allocated to either the training or test set. In the *Distance Split*, we enforce specific median distances between the training and test peptides, using either sequence distance (BLOSUM) or shape distance (RMSD). The training-test distance between peptides is controlled by selecting three percentile intervals over the cumulative median peptide-peptide distance distribution: (0,33), (33,66) and (66,100). Mean and 95% confidence intervals are calculated over 5 repeated experiments with different random seeds. More detailed results are available in S1 File Sect 2, with Levenshtein DS (S15 Fig), overall and per-peptide metrics (S1 File Sects 2.2.2 and 2.2.3).

We divide the peptides into different training and test sets using a distance-based algorithm using a sequence metric (BLOSUM) and a shape metric (RMSD). To obtain the RMSD, we generate the 3D structures of the antigenic peptides using ESMFold and OmegaFold [37, 38]. We then use PyMol to align and calculate the Root Mean Squared Distance (RMSD) between the Cα atoms in the protein backbone [34]. We use Cα RMSD, as opposed to all-atoms RMSD, as it is computationally less intensive and produces similar results [43]. Additionally, predicted 3D structures can (and often will) have artifacts such as physically-unlikely positioning of the amino acid side chains, which could unfairly increase the all-atom RMSD [41].

To simulate real-world scenarios involving unseen peptides in a controlled manner, we propose the *Distance Split* (DS) approach (see Algorithm 1). Using the DS, we chose each test split to guarantee a given median distance between peptides in the training and test peptides. By using the median, some peptides with properties similar to those in the training set may still be included in the test set, thereby allowing for a degree of overlap while maintaining sufficient variability. Nevertheless, when employing the DS, analogously to the HS, test and validation peptides cannot appear at training time. For more stringent conditions, the minimum distance could be used with the DS to ensure that all test peptides are distinctly different from those in the training set.

Specifically, the test peptides are chosen by sampling from a subset of peptides, whose median distance from the other peptides is between a lower and an upper bound. In our experiments, we consider the following intervals over the cumulative median distance distribution: (0,33), (33,66) and (66,100).

As shown in Fig 2, we observe no correlation between RMSD and BLOSUM. Several factors could explain this. The 3D structure, as opposed to the amino acid sequence, is more relevant to describe protein-protein interactions. For example, certain angles in structures could make specific areas interact more than others. Related to this, some small differences

in sequence metrics may result in larger structural differences, for instance, if two oppositely charged amino acids are placed next to each other. The 3D structure and RMSD metric can therefore provide a stronger signal for binding prediction models and may be used for testing models performance in out-of-distribution settings, or for the development of new models. Interestingly, we observe that models' performance is slightly lower for DS in the ranges (33,66) and (66,100) compared to that of the HS. The HS ensures that both test and training peptides are *unique*. However, even with this uniqueness, test and training peptides might still possess a certain degree of similarity. This similarity provides the model with additional chemical information about the peptide. In contrast, DS enforces a threshold distance between the training and test sets, thereby controlling the difficulty.

Obtaining a 3D structure of a protein often requires lengthy experiments to crystallize it, which may take days, months, or years [44]. With the advent of sequence-to-shape models like AlphaFold [45], OmegaFold [38] and ESMFold [37], this lengthy process can often be reduced to minutes or hours and makes it possible to obtain a fairly accurate 3D shapes of proteins. Structural validation in the lab of these models is currently underway, with some researchers reporting partial success, even when solving for previously unknown structures [46]. Successful applications of sequence-to-shape models in protein research include vaccine design [47], binding affinity ranking [48], protein sequence design [49,50] and benchmarking [51].

The structures produced by sequence-to-shape models like the ones mentioned above, may be biased towards the training sequences in the Protein Data Bank (PDB), which is composed of larger structures compared to peptides, which are made up of just a few amino acids in length [52]. Nevertheless, in this study, we use the shape as a representation of the real shape and assume that differences between two shapes would be consistent. This analysis could be extended to the full VDJdb dataset to test how the findings transfer to non-viral peptides.

In the structure-based DS, as the median RMSD distance between training and test peptide is increased, the model performance generally worsens. On the other hand, we observe the opposite effect in sequence-based DS, where increasing the BLOSUM sequence between training-test leads to better generalization performance (See Fig 3).

The immune system has evolved to recognize multitudes of pathogens, some of which may have never existed. The TCR-pMHC interaction is therefore degenerate, meaning that many TCRs can recognize the same peptide and that a TCR can recognize millions of peptides, even if the sequences are not identical [53]. Additionally, TCRs may bind different peptides using different binding modes [54]. However, as shown in Fig 2, we find no correlation between sequence and shape distance metrics *for the peptide*. Coupled with the results of Fig 3, this may imply that TCRs bind conformationally similar viral peptides even if the sequences are different. This is supported by the fact that the peptide binding pocket of MHCs have anchor positions, like the B pocket, which can accommodate different secondary anchor amino acids without altering the overall peptide conformation [55,56]. This flexibility allows TCRs to recognize structurally similar peptides despite sequence variations, suggesting that structural rather than purely sequence-based features drive TCR recognition efficiency in some cases [55,56]. Additionally, viruses, like SARS-CoV-2, evolve by changing their sequences to escape immune detection while keeping the structural features needed to interact with host cells. For example, mutations in the spike protein allow SARS-CoV-2 to evade the immune system but still bind effectively to the ACE2 receptor, ensuring the virus remains functional [57,58]. This suggests that TCRs may bind structurally similar peptides even when their sequences differ, which could explain why models trained on sequence-diverse data (high BLOSUM distance) show improved generalization. In the BLOSUM (0,33) split, where training peptides share

higher sequence similarity, models may overfit to sequence-based patterns, potentially memorizing sequence motifs rather than learning underlying structural determinants of binding, leading to lower test performance. This may explain why models trained on more sequence-diverse peptides (33,66) and (66,100) generalize better, as they are forced to rely on broader structural and contextual features.

Additionally, in the field of protein design, benchmarking has revealed that even with a sequence similarity below 50%, the predicted 3D structures look very similar [51]. These results suggest that BLOSUM-based sequence augmentation, aimed to generate similar binding peptides for a given TCR, may reduce generalization. In contrast, a shape-based augmentation, for example by using protein sequence design models to generate new peptides, may enhance generalization to unseen peptides [49,59]. Additionally, given that the peptide sequence and shape are not correlated, including the 3D shape may increase the overall performance, as shown by recent papers [60–63] However, as TCR and MHC structures are generally similar, it may be worthwhile to limit the 3D information to the regions that vary in shape, like the peptide, the CDRs, and the binding pocket to reduce the model complexity and maintain performance [62]. Future work can explore the differences in structural distance between models such as AlphaFold, ESMFold and OmegaFold in the context of peptides, to further evaluate biases and divergences of structures generated by different sequence-to-shape models.

A short-coming of the BLOSUM distance for the peptide is that changes in the anchor amino acids should be weighted more than changes outside the anchors. Future work could explore a more peptide-centric metric of distance based on anchor positions. Similarly, a short-coming of the RMSD is that due to differences in peptide lengths, it is not possible to normalize the distance by the length of the peptide. In our pairwise comparison, however, most (about 87%)[2] of the peptide distances were same-length comparison. Furthermore, the RMSD is computed on predicted structures, which may contain modeling artifacts. These artifacts could introduce noise in structure-based splits, potentially leading to misleading structure-based generalization patterns. Future work could explore the RMSD of the core amino acids of the peptide and using length-normalized RMSD like $RMSD_{100}$ [64].

Additionally, datasets with highly clustered peptides may present disproportionately low median distances, leading to unintended effects where the test set consists largely of local clusters rather than globally diverse peptides. Conversely, outliers with high median distances will always be placed in test or validation, making these sets more challenging than intended. We mitigate this by enforcing minimum and maximum peptide counts. Future work could explore clustering-based analysis of the peptide datasets or clustering to produce balanced peptide representation in training-test splits.

Despite the limitations of both sequence- and structure-based metrics, the DS algorithm is designed to be metric-agnostic, meaning it can be applied to any distance function that quantifies peptide similarity (see S1 File Sect 2.2.4 for results obtained using the Min operation instead of Mean in the DS algorithm). However, the choice of distance metric significantly impacts the nature of the training-test split and, consequently, model generalization. Future work could explore more biologically or immunologically relevant distances, such as a BLOSUM matrix derived from TCR-pMHC interaction data or alternative physico-chemical distance metrics [65].

Sequence-to-shape models could also be used to predict full TCR complexes for binding prediction. However, current methods for structural complex modeling are computationally expensive. While they may be comparably faster than molecular dynamics simulations,

---

[2]  248,630 matching length pairs comparisons out of 285,090 using 757 peptides

it is currently unfeasible to test all possible combinations of TCRs, peptides and pMHCs for the binding prediction task. Future work in lightweight representations for proteins, such as fragments, could help bridge this gap, allowing for faster and more efficient incorporation of structural features in TCR binding prediction [66].

## Conclusion

Our results suggest that the use of 3D shapes in the context of TCR-pMHC interaction prediction could help reduce the uncertainty about the generalization capabilities of ML models to unseen sequences. Given enough computational resources, 3D shapes could be predicted for the whole TCR structure, as well as for the MHCs presenting the peptide. This would enable the design of models that, given the individual structures of each input sequence, will predict more accurate binding interactions.

## Supporting information

**S1 File. Supplementary Tables and Figures.** Supplementary figures and tables referenced in the main text, including TCR-pMHC complex diagrams, additional model results, and extended correlation tables. See also Supplementary Sects 2.1–2.3 for details.
(PDF)

## Author contributions

**Conceptualization:** Leonardo V. Castorina, Filippo Grazioli, Pierre Machart, Anja Mösch, Federico Errica.

**Investigation:** Leonardo V. Castorina, Filippo Grazioli, Pierre Machart, Federico Errica.

**Methodology:** Leonardo V. Castorina, Filippo Grazioli, Pierre Machart, Federico Errica.

**Project administration:** Leonardo V. Castorina, Filippo Grazioli, Pierre Machart.

**Resources:** Leonardo V. Castorina, Filippo Grazioli.

**Software:** Leonardo V. Castorina, Filippo Grazioli.

**Supervision:** Filippo Grazioli, Pierre Machart, Anja Mösch, Federico Errica.

**Validation:** Leonardo V. Castorina, Filippo Grazioli.

**Visualization:** Leonardo V. Castorina, Filippo Grazioli.

**Writing – original draft:** Leonardo V. Castorina, Filippo Grazioli, Pierre Machart, Anja Mösch, Federico Errica.

**Writing – review & editing:** Leonardo V. Castorina, Filippo Grazioli, Pierre Machart, Anja Mösch, Federico Errica.

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
