## [Decision Letter · Decision Letter 0]

28 May 2024

PONE-D-24-11166Assessing the Generalization Capabilities of TCR Binding Predictors via Peptide Distance AnalysisPLOS ONE

Dear Dr. Castorina,

Thank you for submitting your manuscript to PLOS ONE. After careful consideration, we feel that it has merit but does not fully meet PLOS ONE’s publication criteria as it currently stands. Therefore, we invite you to submit a revised version of the manuscript that addresses the points raised during the review process.

We look forward to receiving your revised manuscript.

Kind regards,

Amit Kumar

Academic Editor

PLOS ONE

Journal Requirements:

"Funding for this project was provided by NEC Labs Europe GmbH."

"Funding for this project was provided by NEC Labs Europe GmbH."

Reviewers' comments:

Reviewer's Responses to Questions

**Comments to the Author**

1. Is the manuscript technically sound, and do the data support the conclusions?

Reviewer #1: Partly

Reviewer #2: Partly

2. Has the statistical analysis been performed appropriately and rigorously? 

Reviewer #1: No

Reviewer #2: Yes

3. Have the authors made all data underlying the findings in their manuscript fully available?

Reviewer #1: Yes

Reviewer #2: Yes

4. Is the manuscript presented in an intelligible fashion and written in standard English?

Reviewer #1: Yes

Reviewer #2: Yes

5. Review Comments to the Author

Reviewer #1: The authors introduce a novel approach to assess the generalization capabilities of TCR-pMHC binding predictors through the Distance Split (DS) method. This technique evaluates machine learning (ML) models, such as the Attentive Variational Information Bottleneck (AVIB) and NetTCR-2.0, using both sequence-based and structural (3D) metrics. They focus on the importance of structural metrics, leveraging advances in protein modeling with third-party tools like ESMFold and OmegaFold to enhance the prediction accuracy of ML models. This study emphasizes the necessity for robust evaluation methods that consider 3D structural distances to predict the real-world applicability of TCR-pMHC interaction models more reliably.

The motivation and objective of the research are founded on a solid and clear background, making the concept of the paper highly comprehensible and appreciable. However, there are several aspects of the methodology, particularly the new split method, which could benefit from further clarification and justification. The framework for justifying this new method seems to require more robust reasoning to convincingly support the claims made. Given these concerns, I would suggest substantial revisions to address these points before reconsidering the paper for publication.

[Major comments]

1. Dataset Split Evaluation

The paper discusses the relationship between performance decline and distance magnitude when using the DS algorithm with structural distance. It should be more insightful to include comparisons across models trained on different splits, such as DS versus HS, and evaluated on a common ground truth test set that is created via DS or HS. In short, when stating the usefulness of the dataset split based on the structural distance, it is not enough to show only the performance decline versus the distance magnitude, probably.

This would provide a clearer picture of the effectiveness of each splitting method when tackling a real-world scenario, where researchers train models on whatever the split is and use the models on the unknown test set which might or might not be structurally similar to the training set.

2. Clarification of DS Algorithm Description

The DS algorithm's definition and explanation would benefit from being included in the main body of the paper rather than the Supplementary Materials because it is the main part of the research.

A more detailed description could address potential ambiguities, such as whether the use of median distance might still allow for overlap in peptide characteristics between the training and test sets. I mean, if it takes the median of the distance per peptide when setting a test set, does it still contain a chance of having similar peptide set, similar with training? Perhaps considering the minimum distance per peptide for splitting would offer a stricter and potentially more effective division after creating the vector m for each peptide.

The methodology for sampling peptides for the test and validation sets raises concerns regarding potential overfitting, especially because the validation data seems too similar to the test data. Clarifying the strategy behind this choice and considering the implications of such a setup in real-world scenarios would be beneficial.

What is a leeway factor "l" and what is the budget and what is the sN of N_test relationship of them?

3. Are the two benchmark models based on the sequence?

If so, it was remarkable to observe that the longer distance split did not change the AUC.

If there is a way to include structural features in peptide, do you think the longer distance of RMSD did not affect the AUC?

4. Peptide distance

The method of calculating distances might be impacted by peptide length variations. Including information on peptide lengths and discussing how they affect distance calculations would provide a more thorough understanding of the model's applicability.

Additionally, is alpha carbon enough to model the peptide-peptide similarity? Often the case, side chains of amino acid form the bond with other molecules.

[Minor comments]

* In evaluating TCR-pMHC these papers are good to cite, https://doi.org/10.3389/fimmu.2020.01803, https://doi.org/10.3389/fbinf.2023.1274599, 10.1186/s12859-019-3109-6

* Line9: what does this mean by the word "on ML model risk assessments" in the Abstract?

* Line 176: the citation of the benchmark model is inconsistent. "Jurtz et al. (2018)" and "(Grazioli et al., 2022b)". The end of sentence is unclear.

* Line 177: "This model presents probabilistic encoders". Which model is it?

* In Table 1, the average of peptide counts is 45, but why is it not "87.1 = 65,946 / 757"

* Line 200: why are there 5 different random seeds? Is there any randomness in DS splits?

* Do some of the metrics need thresholds of the confidence values in Figure S2?

* There is a missing period in the caption of Figure 4.

Reviewer #2: The manuscript presents an interesting and important aspect of unseen epitope binding prediction (to TCRs): structure based train val test splitting instead of unique or sequence similarity based. The authors show that the the structural distance could be used better than sequence based distances for uncertainty estimates. However, the current manuscript is not thorough enough. The current way of presenting does not present distance based uncertainty (which would be very valuable i.e. this peptide is too distance the model cannot evaluate the binding probability, also quite hard) of a new peptide rather than generally structural differences are more important than sequence identity.

Major

-More (diverse) methods in the comparison

There are only two methods presented and they use the same amino acid representation and similar architecture. To add more trustworthiness for the claims it would be important to add more methods that use different architectures and input representations. (Also NetTCR2 is used instead of the newer version NetTCR2.2)

-What is the importance of the new splitting why not use unseen epitope test (HS) and then bin by distances like done in e.g. ImRex, EPIC-TRACE, MixTCRpred

An addition of epitope score binning (and correlation) would be a valuable addition to verify the results. (also by using different binning methods on HS many methods can be analyzed with same computational cost of training the models/ or more seeds can be evaluated)

-The DS seems to control only for datapoint number but not number of peptides, which feels quite important in this case as the test amount can be filled with 1 or with many peptides

-Did you try DS with min distance, or mean, or mean of min X e.g. 10? What is the reasoning behind the median?

-So What are the advantages of splitting versus binning on HS?

Minor

-Add String edit distance (Levenshtein) as a simple baseline

-and maybe some physicochemical property distance or other distance that is different from sequence and structure

Data is gathered from public database.

- Regarding Figure 1/2 the most abundant epitopes are not shown in the figures as the distribution of datapoints per epitope is unbalanced and some epitopes have thousands of datapoints whereas most have only a few. This is somewhat confusing for the interpretability of the plot

-Also why is the mean plotted while the median is used in in the split?

- I believe that the performance should be for peptides not for datapoints thus choosing macro (separately for each peptide and then average) over micro (over all datapoints directly) AUROC could be considered

-Number of unique test peptides (mean) in each split would also be informative (even more so when binning HS and/or using macro AUROC)

Also, check this out. Cannot require the comparison to a method that was published after submission, but this paper seems to have same direction results, improving prediction performance by using the epitope structure:

"Hongchen Ji, Xiang-Xu Wang, Qiong Zhang, Chengkai Zhang, Hong-Mei Zhang, Predicting TCR sequences for unseen antigen epitopes using structural and sequence features, Briefings in Bioinformatics, Volume 25, Issue 3, May 2024, bbae210, https://doi.org/10.1093/bib/bbae210"

6. PLOS authors have the option to publish the peer review history of their article (what does this mean?). If published, this will include your full peer review and any attached files.

Reviewer #1: **Yes: **Kyohei Koyama

Reviewer #2: No

---

## [Author Response · Author response to Decision Letter 1]

21 Oct 2024

Dear Reviewers,

We would like to express our sincere gratitude for your thoughtful and constructive feedback on our manuscript. We have carefully considered all the reviewers’ comments and have made revisions to address each point. Generally, we expanded the paper to have more TCR-pMHC models to ensure the robustness of results. Additionally, we removed LocalDS because the way it was implemented was not weighing the score by BLOSUM weights. Instead, we added Levenshtein distance as a baseline, and BLOSUM score (obtained after an alignment) to add more physico-chemical information.

Below, we provide detailed responses to each of the comments and explain the corresponding changes made in the manuscript.

We have modified the paper accordingly tracking the changes in purple in the file _NEC__Distance_Split.pdf and replied to each reviewer comment in details in the file _NEC__Distance_Split_letter.pdf

---

## [Decision Letter · Decision Letter 1]

4 Feb 2025

PONE-D-24-11166R1Assessing the Generalization Capabilities of TCR Binding Predictors via Peptide Distance AnalysisPLOS ONE

Dear Dr. Castorina,

Thank you for submitting your manuscript to PLOS ONE. After careful consideration, we feel that it has merit but does not fully meet PLOS ONE’s publication criteria as it currently stands. Therefore, we invite you to submit a revised version of the manuscript that addresses the points raised during the review process.

We look forward to receiving your revised manuscript.

Kind regards,

Amit Kumar

Academic Editor

PLOS ONE

Additional Editor Comments:

Dear Authors,

I sincerely apologize for the delay in reviewing your manuscript. Unfortunately, only one of the two invited reviewers responded, which caused a delay.

Please address the comment and submit a revised version.

We appreciate your patience and understanding.

Reviewers' comments:

Reviewer's Responses to Questions

**Comments to the Author**

1. If the authors have adequately addressed your comments raised in a previous round of review and you feel that this manuscript is now acceptable for publication, you may indicate that here to bypass the “Comments to the Author” section, enter your conflict of interest statement in the “Confidential to Editor” section, and submit your "Accept" recommendation.

Reviewer #2: (No Response)

2. Is the manuscript technically sound, and do the data support the conclusions?

Reviewer #2: Partly

3. Has the statistical analysis been performed appropriately and rigorously? 

Reviewer #2: Yes

4. Have the authors made all data underlying the findings in their manuscript fully available?

Reviewer #2: Yes

5. Is the manuscript presented in an intelligible fashion and written in standard English?

Reviewer #2: Yes

6. Review Comments to the Author

Reviewer #2: Many of the concerns raised during first review round are applied. The algorithm is better presented and it is used for many recent methods. The work still aims to tackle an important issue and it seems that some method of splitting with a controllable similarity will be important for/in the future. However currently the biases/unknowns of the method seems a bit unclear.

Good that BLOSUM and Levenshtein baselines were added.

Interesting to see the macro AUROCS telling quite a different story than Micro.

Nice that the number of peptides seems to be split reasonably.

SUMMARY:

The hearth of the paper is DS splitting. Thus the method needs to be understood, including its potential limitations and biases. Also it needs to be shown relevant i.e. that a simple HS split with binning/distance based analysis on the random HS test split is not sufficient now or in the future (e.g. when more peptides are available). The importance of structure is highlighted which a strong point and is in line with some recent work. Thus by clarifying the potential limitations and showing and clarifying the importance over only HS and post splitting analysis would make the methodology more rigorous.

Major

1) What are the biases of the method? What does it assume? Are there any practical considerations?

BLOSUM has inverse relation which is hypothesized to cause of more diverse training (lines 127-128). This seems very interesting and could be discussed more. Also is there any supporting results for this (It is not clear what the suppl Entropy tables are if related to this statement)

Median is calculated for the full set of unique peptides (to be split into train,val,test) thinking of a failure case. If there is a big cluster of peptides with all dist 1 of each other (this is not entirely unlikely e.g. TULIP dataset contains all 1 aa mutations of the original) then the cluster would have a big impact in lowering the median score of all peptides in the cluster but the distance to other peptides outside the cluster would be larger an quite constant, thus choosing the lower percentile (including mostly the cluster) this should according to the idea of the splitting have a low distance to the train but now the distance is actually higher as most of the low distance contribution is in the cluster that is set to val test.

Also the peptides with high median are probably "outliers" meaning they have higher distances to all other (no clustering) thus these peptides would in any case be longer distance of the train, read given a random HS split on some of these "outlier" peptides will always be in the test/val. If this some amount is sufficiently large then the DS splitting seems less useful.

How does the data (skewed) distribution affect the sampling results?

2) Please try binning on HS (if that was not already done?)

A comparison against HS with binning would still be needed.

In the future when there is a massive (hopefully we will come to this) amount of peptides the author is correct a random (HS) split might well only have close peptides and then the evaluation must be split. Also in this case I would recommend binning and correlation between actual distance and performance to get the evaluation.

this could be the future approach: splitting to ensure the approx distance to train + binning and correlation for evaluation to see more fine-grained distance.performance relation.

What are the correlations in Figures 3 and S13 and Tables S1-4. Correlations between peptide distance and performance on HS or on distance splits (combined) ? also what is micro correlation (these needs to be binned so what are the bins the thse DS splits or arbitrary smaller bins)

Optimally I would prefer both of the peptide-performance correlations and only barplots/ tables would be fine for binns.

Middle

1) Both reviewers asked about minimum dist split, it would have been nice to see how it performs.

(also response 4.1 the median/ mean or anything is calculated in the space of unique peptides right thus the unbalance would not be affecting the metric (any metric would probably be very ill suited if calculated on the datapoint space vs unique peptide as some 20% probably is of the most frequent peptide))

Minor

1) what are the 5 - 5000 line

What does this mean does it force the test/val to have between 5-5000 unique peptides or force the test/val to have only peptides with 5-5000 datapoints. (in essence excluding the most frequent always from test/val)

2) Figures and Tables in supplementary are not noted/discussed/referenced anywhere, adding a mention from the main text and analysis/conclusion (either main or suppl would be good)

3) Levenshtein median seems very ill suited for this the average or minimum might be better

4) previous minor 2) the figure could use logarithmic binning on the y axis, accounting for all peptides in non-log space is obviously not sensible as the authors also show. On one hand it is true that the marginals would be harder to interpret on the other hand the most frequent are very much missing from illustration(s), which are definitely very important for the analysis. I agree that this is hard to visualize maybe showing the non-log space and setting x-axis marks for the most frequent ones, if this plotting is feasible for visual appeal. Also and answer to Minor 1, if most frequent are always in train then they are not so interesting in what median distance pocket it is in.

5) line 99 maybe cite ERGO here

6) Usually the peptide is rotated such that the anchor position amino acids of the peptide are facing the MHC and the rest maybe then facing the TCR can this be seen from the alpha carbon positions?

7) line 246 hints that hard split was invented in Grazioli 2022 although the unseen epitope/strict/TPP3orTPP4 splits have been used earlier

8)line 317 318 "It is therefore..." I do not follow the logic and the statement remains unclear but feels like an important hypothesis, could you please clarify.

7. PLOS authors have the option to publish the peer review history of their article (what does this mean?). If published, this will include your full peer review and any attached files.

Reviewer #2: No

---

## [Author Response · Author response to Decision Letter 2]

4 Apr 2025

Dear Dr. Magallanes, dear Reviewers,

We sincerely appreciate your valuable and insightful feedback on our manuscript. We have thoroughly reviewed all the review comments and have made the necessary revisions to address each point accord-

ingly.

We provide detailed responses to each of the comments and explain the corresponding changes made in the manuscript in the cover letter file

Sincerely,

The authors

---

## [Editor Report · Decision Letter 2]

20 Apr 2025

Assessing the Generalization Capabilities of TCR Binding Predictors via Peptide Distance Analysis

PONE-D-24-11166R2

Dear Dr. Castorina,

We’re pleased to inform you that your manuscript has been judged scientifically suitable for publication and will be formally accepted for publication once it meets all outstanding technical requirements.

Kind regards,

Amit Kumar

Academic Editor

PLOS ONE

Additional Editor Comments (optional):

The paper can be considered for publication.
---

## [Editor Report · Acceptance letter]

PONE-D-24-11166R2

PLOS ONE

Dear Dr. Castorina,

I'm pleased to inform you that your manuscript has been deemed suitable for publication in PLOS ONE. Congratulations! Your manuscript is now being handed over to our production team.

Kind regards,

on behalf of

Dr. Amit Kumar

Academic Editor

PLOS ONE